# Functional Importance of the Hydrophobic Residue 362 in Influenza A PB1 Subunit

**DOI:** 10.3390/v15020396

**Published:** 2023-01-30

**Authors:** Johnson Jor-Shing Chan, Yun-Sang Tang, Chun-Yeung Lo, Pang-Chui Shaw

**Affiliations:** 1Centre of Protein Science and Crystallography, School of Life Sciences, The Chinese University of Hong Kong, Hong Kong, China; 2Li Dak Sum Yip Yio Chin R&D Centre for Chinese Medicine, The Chinese University of Hong Kong, Hong Kong, China

**Keywords:** influenza, RNA-dependent RNA polymerase, PB1 β-hairpin

## Abstract

PB1, acting as the catalytic subunit of the influenza polymerase, has numerous sequentially and structurally conserved regions. It has been observed that the slight modification of residues in PB1 would greatly affect the polymerase activity and even host adaptation ability. Here, we identified a critical residue, 362M, on the polymerase activity and virus replication. By means of the minireplicon assay, we assured the importance of the hydrophobicity of PB1 362, and the possibility that the size and charge of the side chain might directly interfere with the polymerase function. We also proposed a hydrophobic core between the PA-arch and the PB1 β-hairpin motifs and showed the importance of the core to the polymerase function.

## 1. Introduction

Influenza A virus (IAV) is a segmented negative-sense RNA virus, possessing eight independent gene segments encoding at least 13 viral proteins. Viral genomes are assembled as the viral ribonucleoprotein (vRNP) in the virion, where the 5′ and 3′ termini are embedded in the polymerase head in a partial-complementary manner, and the remaining portion is stabilized by the NP polymer helix [1,2,3]. The heterotrimeric polymerase, composed of the polymerase basic subunits 1, 2 and polymerase acidic subunit (PB1, PB2 and PA), is critical for genomic replication and transcription upon infection. The PB1, PB2 N-terminal and PA C-terminal domains form the conserved polymerase core, while the remaining parts of PB2 and PA are responsible for additional functions of the polymerase core [4]. For example, the PA endonuclease and PB2 cap-binding domain are involved in the cap-snatching mechanism [3,5], generating primers from the host pre-mRNA for the primer-dependent transcription initiation. The polymerase core uptakes the template, recruits rNTPs and synthesizes different RNA through different pathways [6,7]. The viral RNA polymerase makes use of the 5′-capped primer to transcribe mRNA with both capped 5′-end and poly(A) tails. Meanwhile, the polymerase replicates the viral RNA to generate a complementary RNA (cRNA) without cap and adenylation. The cRNA acts as a replicative intermediate for the synthesis of vRNA for assembly into new virions [8].

In 2014, the crystal structure of IAV polymerase was solved [9] and different domains were mapped. As the polymerase core and catalytic subunit, PB1 has a well-known right-handed fold with the typical A-E and pre-A motifs of RNA-dependent RNA polymerases. PB1 protein is structurally and sequentially conserved evolutionarily, despite some mutations and reassortments being identified [10], which might be implicated in host specificity and pandemic potential. A recent study has captured the dynamic snapshots of the active polymerase transcribing an mRNA [11], showcasing the important conformational changes on the different polymerase subunits and how they are structurally coordinated during transcription.

The PB1 β-hairpin located between motif A and B is the key promoter binding region, interacting with the 5′ promoter hook of the vRNA promoter [3,9] and the PA arch motif. Conserved amino acids on the two β sheets interact directly and stabilize the RNA backbone, in which residues 356-Met-Phe-Glu-358 and Arg365 are critically important in the interaction. Surveying available influenza polymerase sequences and structures from the databank, we observed a β turn with the sequence 359-S-K-S/R-M-K/R-L-364 present at the hairpin tip is conserved among IAV strains. Alignment of complete PB1 protein sequences in the Flu Databank (https://archive.ph/roo0m (accessed on 5 January 2023)) suggested an extremely conserved residue PB1 362M at the hairpin tip, with rare natural variants of PB1 362M in IAV including 362I, 362L and 362V, leading to the hypothesis on the importance of a hydrophobic residue at position 362 in IAV PB1.

In this study, we demonstrate that hydrophobic residue 362M is crucial for polymerase function and viral fitness. We found that the mutation M362A drastically reduced polymerase activity, while M362L significantly increased the replication and transcription efficiency. Our study highlighted the functional importance of PB1 residue 362 in IAV and provided molecular insights into how PB1 hairpin interacts with the PA arch to fulfill its structural functions.

## 2. Material and Methods

### 2.1. Plasmids Constructs and Mutagenesis

All four subunits comprising the RNP of both A/WSN/1933 and A/Hong Kong/156/97 were cloned into the mammalian expression vector pcDNA3 for cell-based expression and activity assays. Mutants of PB1 were generated by site-directed mutagenesis and sequenced for validation.

For the assessment of polymerase activity, reporter plasmid was required to be co-transfected with the RNP plasmids. WSN-NA and firefly luciferase reporter plasmids with human Pol-I promoter flanking the influenza 5′ UTR [12] were kindly provided by Prof. Ervin Fodor (Sir William Dunn School of Pathology, University of Oxford).

For the generation of recombinant viruses, all eight segments of A/WSN/1933 were constituted into pHW2000 backbone [13] and the plasmids were kindly provided by Dr. Chris Mok (Jockey Club School of Public Health and Primary Care, CUHK). Mutants of PB1 were generated by site-directed mutagenesis with the same primers as mentioned (Appendix A) and cloned plasmids were sequenced for validation.

### 2.2. Sequence Analysis of Influenza Virus

A total of 56,448 complete sequences of IAV PB1 genes were retrieved from Influenza Research Database ranging from before 1980 to 2020. Redundant sequences were removed. The sequences were then compared and analyzed through BioEdit [14] and sequence alignment program in RStudio. Sequences were sorted and aligned by the β-hairpin with reference to the structures available on RCSB (RCSB ID: 4WSB, 6RR7) [9,15]. Sequences were then sorted by year of isolation and subtypes.

### 2.3. Minireplicon Assay

Human embryonic kidney (HEK) 293T cells (293T) were resuspended, transfected with viral RNP expression plasmids (i.e., PB2, PA, NP, wild-type/mutant PB1, 0.125 μg each) together with firefly luciferase reporter plasmid (0.125 μg). EGFP expression plasmid pEGFP (0.0625 μg) was together co-transfected in the assay as the internal control to normalize the transfection efficiency. D445A-D446A PB1 deficiency mutant [16] was used as negative control of the assay. In-suspension transfections were performed by Lipofectamine2000 reagent according to manufacturer’s instruction. Each reaction was performed in a single well in a 96-well plate.

At 48 hours post-transfection, cells were lysed by Luciferase Lysis Buffer (25 mM Tris-Phosphate, 2 mM DTT, 10% Glycerol, 1% NP-40, pH7.8) and fluorescence signal was detected by BMG CLARIOstar microplate reader. Lysates were then transferred to OptiPlate-96 and Steady-Glo luciferase substrate was added in 1:3 ratio of the lysate and the luminescence signal was recorded by a plate reader. The remaining lysate was subjected to Western blot analysis. The luciferase activity level was normalized with that from the wild-type PB1.

### 2.4. RNA Quantification and Identification by qPCR

293T cells were added to a 12-well plate one day prior to transfection. Plasmids expressing viral RNP, PB1 mutants and firefly luciferase reporter as mentioned above were together co-transfected by polyethylenime (PEI) in a ratio of 1:2.5.

Cells were washed and resuspended with PBS at 24 hours post-transfection. 1/10 of the transfected cells were subjected to Western Blot analysis while the remaining portion was lysed by RNAiso Plus (TaKaRa) reagent, followed by chloroform-isopropanol RNA isolation procedures. Samples were then subjected to DNAse treatment by amplification grade DNAse I (Thermofisher, Waltham, MA, USA) and concentration of the treated RNA was determined by spectrophotometry.

Hot start RT-qPCR was performed with slight modifications [17]. In brief, specifically designed primers with tag at the 5′ end were used in reverse transcription by M-MLV reverse transcriptase under 50 °C incubation. Samples were then amplified by Power SYBR^®^ Green mastermix and analyzed by 7500-fast real time PCR machine (ABI).

### 2.5. Antibodies and Western Blot Analysis

Samples in each assay were analyzed by Western Blot analysis for the validation of protein identity. Sampling buffer was added and samples heated to 98 °C for 10 min before being analyzed by SDS-8% PAGE.

Proteins transferred to PVDF membrane were probed by antibodies targeting PB1, PB2 and PA, respectively. Antibodies targeting PB2 and PA were purchased through GeneTex, while antibody targeting PB1 was obtained from BEI Resources (Catalog number: NR-31691). Bound antibodies were detected by using HRP-conjugated anti-mouse or anti-rabbit IgG secondary antibody (Bio-Rad, Hercules, CA, USA) and ECL development (GE Healthcare, Chicago, IL, USA) according to the manufacturer’s instructions.

### 2.6. Recombinant Virus Reconstitution

Recombinant viruses were generated in vitro by reverse genetics, as previously described [18] with slight modifications. In brief, 293T cells and Madin-Darby canine kidney (MDCK) cells were plated onto 6-cm dishes in 2:1 ratio. The cells were co-transfected with the eight pHW2000-based plasmids (2 μg each) using PEI in 2:1 ratio.

After 18 hours post-transfection for 18 h, cells were washed with pre-warmed PBS and serum-containing medium was changed into serum-free Minimal Essential Medium (MEM) containing 0.1% TPCK-treated trypsin. Supernatants were collected after 96 h and transferred to new MDCK culture for viral recovery. The recombinant viruses were amplified by 3–5 passages before virus titer determination was performed. Total RNA of infected cells was harvested and RT-PCR was performed, followed by Sanger sequencing to confirm the mutations.

### 2.7. Viral Growth Curve

MDCK cells were plated at 5.0 × 10^5^ cells/well in 6-well plates before infection. Cells were washed twice with PBS prior to the infection at a MOI of 0.005. Supernatant was collected at indicated hours post-infection. Viral titer of each inoculate was determined by plaque assay [19].

### 2.8. Statistical Analysis

Data represent at least three independent biological replicates. Quantitative data are shown as mean ± standard error of measurement for multiple biological replicates. Unless otherwise specified, data comparisons were performed by two-tailed Student’s *t*-test.

## 3. Results

### 3.1. PB1_362_ in IAV Is highly Conserved among Different Strains 

We evaluated PB1 sequences of influenza A strains available in the Influenza Research Database. Sequence alignment suggested an extremely conserved residue at the tip of the β-hairpin, which is a Met for 99.67% of entries (Appendix A). Other than Met, we identified several natural variations at this position. The most frequently observed variant was M362L, which was found in 15 swine, 17 avian, 1 canine and 1 human virus PB1 protein. Some other hydrophobic residues were also observed at this position.

PB1 sequences in the database were aligned according to the year of isolation: Before 1980, 1980–2000, 2000–2010 and 2010–2020. IAV strains from avian and swine showed greater variability at residue 362 of PB1 subunit, while human strains had highly conserved Met at the same position (Appendix A).

Among all isolates, IAV PB1_362_ was mainly found to be hydrophobic in nature with only two exceptions throughout the 56,448 sequences in the database.

### 3.2. Hydrophobic Residue on PB1 Position 362 Is Critical for Polymerase Activity

As the sequence survey suggested, PB1_362_ in IAVs is extremely conserved evolutionarily, with over 99% Met among different strains at the PB1 hairpin motif 359-Ser-Lys-Ser-Met-Lys-Leu-364. Natural variants were also hydrophobic. We then carried out a minireplicon assay with firefly luciferase (FLuc) as a reporter to review the transcription and replication efficiency of the different hydrophobic mutants. 

A/WSN/1933 (H1N1) (WSN) and A/HongKong/156/97 (H5N1) (156/97) were employed as initial models and their corresponding PB1_362_ were mutated into hydrophobic or charged residues. PB1_362_ is an Ile natural variant in WSN but a Met in 156/97, thus allowing further functional comparison by minireplicon assay. The PB1-D445A-D446A catalytically defective mutant was used as the negative control of the assay.

In our study, 362M and 362L in both strains showed the highest polymerase activity among all mutants (Figure 1A,B). In 156/97, 362L had the polymerase activity increased by 1.7-fold, while a 2-fold increment was shown in WSN (Figure 1B). 362M showed the second-highest polymerase activity in both H5N1- and H1N1-based reconstitutions. There was a 1.5-fold increment of reporter signal in the H1N1 strain with I362M substitution.

On the contrary, all the other hydrophobic residues reduced the transcription/replication efficiency significantly. 362V reduced half of the reporter signal compared with wild type in the 156/97 setup, while 362I reduced the activity by about 90% in the same strain. In H1N1 subtype, 362V reduced the signal by around 77% compared with 362I, the reference sequence in the strain. 362A substitution gave the most significant decrement on the transcription and replication efficiency of both strains, which was comparable to the negative control, D445A-D446A.

Charged residues in both strains showed no signal in the reconstitution assay, suggesting that a charge at PB1_362_ would hinder the polymerase activity. The addition of an alanine between position 361 and 362 (Add_A), or deletion of residue 362 (Del_362) at PB1_362_, both knocked off the polymerase activity. These observations suggested the functional and spatial importance of a hydrophobic residue with an extended side chain at PB1_362_. 

### 3.3. PB1-362 Mutants Altered RNA Synthesis

In order to confirm whether mutations at PB1_362_ affect transcription or replication, we performed a quantitative PCR assay to determine how mRNA, vRNA and cRNA syntheses were affected in both 156/97 and WSN settings.

Reconstituted RNPs were co-transfected with reversed luciferase gene flanked by Pol-I promoter as reporter. Total RNA was harvested and reverse transcribed [17], then the products were subjected to qPCR by the amplification of the tag and luciferase segment.

Overall, we observed similar trends of transcription efficiency as observed in the minireplicon assay. In the 156/97 background (Figure 2A), which naturally carries a Met at position 362, we observed a 2.5-fold increment in mRNA expression level for the 362L mutant, while over 90% reduction of mRNA level was detected with 362I mutants. 362A, 362E, 362K together with D445A-D446A negative control, had no significant expression observed in mRNA level. Both deletion of 362M and the addition of an extra alanine at residue 362 showed defective transcription activity. 

The cRNA expression level of the mutants in the 156/97 background showed a similar trend as the mRNA level. A 2.5-fold increased expression level was observed for 362L mutant, while 362V mutant retained at about 30% cRNA expression. The remaining mutants, by comparing them to the D445A-D446A defective mutant, showed the baseline level of cRNA expression. 

In the WSN background (Figure 2B), which naturally carries an Ile at position 362, we observed a similar mRNA expression level of the 362M mutant, and over 1.5-fold increment for the 362L mutant. The remaining mutants 362V, 362A, 362K showed similar transcription activity comparing to the D445A-D446A defective mutant. 

The cRNA expression level of the 362M mutant increased by around 20%, and the 362L mutant revealed a 1.5-fold increment. The remaining mutants showed only baseline cRNA expression level compared to the negative control. To ensure the difference of polymerase activities observed among the mutants was not the result of protein concentration deviation, we performed Western blot analysis to probe for PB1 or its mutants (Figure 2C,D and Appendix A).

### 3.4. Viral Fitness Was Affected by Mutations at PB1-362

Selected variants of PB1_362_ were subjected to viral reconstitution with the pHW2000 WSN background. The only three residues with polymerase activity, 362M, 362I, and 362L, had infectious virions, and plaque assay reported that at least 10^6^ pfu/mL were generated upon the third passage (Figure 3). Infectious virus could not be reconstituted from mutants with low or diminished polymerase activity, including 362V and 362A. 

Growth kinetics of reconstituted viruses were then compared. In brief, MDCK cells were infected with different viruses at MOI 0.001. Supernatant was harvested at the indicated time points after infection, and plaque assay was performed. WSN/362I^(PB1)^ showed a consistently slower growth rate compared with the other two variants, where 10-fold less viral titer was recorded during the first 30 hours and was around five-fold slower at 36 hours and 42 h post-infection. Growth kinetics of WSN/362M^(PB1)^ and WSN/362L^(PB1)^ mainly deviated at early time points. At 12 hours post-infection, WSN/362M^(PB1)^ had a slower growth rate of half a log compared to WSN/362L^(PB1)^.

### 3.5. PB1-362 and PB1-364 Likely Participate in a Hydrophobic Network with Residues on PA Arch

As our results suggested, the hydrophobic nature of PB1_362_ was critical for the polymerase to maintain activity. From available polymerase structures, four hydrophobic residues, 379V, 384C, 387V and 390L, on the PA arch motif with side chains pointing toward PB1_362_ could be identified (Figure 4A) (RCSB ID: 6RR7). Meanwhile, the structure also suggested another hydrophobic residue located on the PB1 hairpin, the PB1-L/I364, is also pointing towards the center of the hydrophobic network. We then hypothesized these surrounding hydrophobic residues may form a hydrophobic network with the PB1_362_. Thus, we investigated whether these residues would interfere with the polymerase function. The four mutants on the PA-arch domain in both 156/97 and WSN were mutated to alanine and tested for activity (Figure 4B,C). V387A and L390A in both strains had the polymerase activity reduced significantly, with V387A having over 80% activity decrement in both strains, while L390A mutants were defective. Both PA-V379A and PA-C384A only mildly affected the polymerase activity. V379A retained about 75% activity in 156/97 and 40% activity in WSN, while PA-C384A had retained over 90% polymerase activity in 156/97 and 60% activity in WSN. Similarly, when both 156/97 and WSN PB1_364_ were mutated to alanine, significant decrement on the polymerase activity was observed. L364A mutant reduced the polymerase activity by 50% in 156/97, while over 90% decrement was observed with the I364A mutation in the WSN RNP cassette (Figure 4D,E).

## 4. Discussion

The high-resolution structures of bat influenza A polymerase [9,20] have led to the mapping of the promoter binding pocket of the 5′ promoter hook. The 5′ hook is sandwiched tightly to the polymerase by β17-18, β20 sheets and the PA-arch-PB1-hairpin pocket [9]. Such interaction network is found to be re-organized during the initiation-elongation transition [11] where the PA and PB1 β-ribbon would rotate from the original position. Human IAV polymerase structure (RCSB ID: 7NJ5) suggested that residues 371-Gly-Glu-Asn-Met-Ala-375 on the PA-arch and residue R365 on PB1 interact with the phosphate backbone of the promoter [9,21,22]. In fact, an extensive hydrophilic interaction network between the PA-arch and the PB1 β-hairpin motifs could be identified, which includes residues 358E, 359S, 361S, 363K, 365R, 380Y, and 381N in PB1 and residues 376P, 377E, 379V, 389D, 391K, and 392Q in PA [9] (Appendix A). However, little information is available on the importance of hydrophobic residues which are sandwiched by a highly polar interaction network, and indeed, of a high degree of conservation. 

As we have shown, the PB1 β-hairpin loop is highly conserved with the sequence ^359^SKSMKL^364^ in IAV (Appendix A). PB1 residue 362 has a conserved hydrophobic nature along the evolutionary timeline, where Met occupies over 99% of the circulates. After the emergence of pdm09, isolates with PB1 362L mutants started to escalate (Appendix A). Although 362M is the dominating sequence in the circulating IAV strains, variants such as 362L and 362I are found in other hosts. To access the functionality of the PB1 mutants, we evaluated the transcription and replication efficiency by minireplicon assay with different naturally existing variants, and mutants with deviating biochemical natures at position 362. Our results in both H5N1 and H1N1 strains suggested the importance of the hydrophobic nature of the residue, which 362M and 362L containing genes result in the highest polymerase activity in the assay. 362I and 362V mutants retained low polymerase activity. On the contrary, 362K, 362E, deletion of residue 362 and the addition of an Ala next to it inhibited the polymerase activity. Other than 362, conserved residue 364 also takes part in the polymerase activity. Swapping of residue 364 into alanine reduced the polymerase activity (Figure 4D,E) by 70% in H5N1 and 90% in H1N1. The minireplicon assay suggested that both hydrophobic residues 362 and 364 in PB1 contributed to the polymerase function.

The RNA synthesis mechanism was further investigated by the qPCR assay, the results of which were consistent with the minireplicon assay. 362M and 362L substitutions reported the highest mRNA, cRNA and vRNA expression level, while 362I and 362V had a significant reduction in both transcription and replication efficiency. Despite the hydrophobic nature, 362A substitution in both assays showed the complete disruption of the polymerase activity.

As observed from the qPCR results, hydrophobic residues like Ile, Met and Leu at position 362 retained the ability of the polymerase to synthesize RNAs in the context of WSN polymerase, while charged residues inactivated the polymerase. A similar case applied to the 156/97 polymerase, except for that Ile suppressed both transcription and replication of the RNP. 

Our data suggested that the polymerase efficiency may be correlated to the size of hydrophobic side chain at position 362. The larger the amino acid occupied, the higher the polymerase activity observed (Figure 2A,B). Structures of IAV polymerases suggest that PB1 362M inserts into the space between PB1 β-hairpin and the PA arch [9,22]. It is likely that the hydrophobic residue interferes with the network that bound the 5′ promoter hook and may eventually affect the conformational stability around the promoter binding pocket.

Interestingly, the same motif on IBV and ICV (RCSB ID: 6QCS and 6Y0C) [11,23], the residues at the tip of the same β-hairpin are hydrophilic in nature, where a lysine is found in IBV and arginine in ICV. The alignment and consensus map (Appendix A) reported the conserved charged nature of the residue at the equivalent positions of IBV and ICV respectively. It is reported that ANP32A is essential for viral replication in human cells and the dimerization site of the ICV polymerase involved the P3 arch and PB1 β- hairpin [23,24], which is different from the dimerization site of IAV polymerase. The clues are pointing to the possibility that the PB1 β-hairpin maybe involved in the interaction between the encapsidating polymerase and the resident polymerase in ICV, where a charged residue at PB1_362_ might be crucial for the interaction. 

Furthermore, our data on the PA arch hydrophobic residues suggested the importance of hydrophobic residues 387V, 390L, and to a less extent, 379V and 384C, to the polymerase activity (Figure 4A,B). Side chains of PA 387V and 390L form a hydrophobic cluster with PB1 362M and PB1 364L. The side chains of these residues are within around 4Å to each other (RCSB ID: 6T0N, 6SZV). As a result, a hydrophobic network should exist between the two motifs. It is possible that the hydrophobic interactions among the suggested residues are compensating the electrostatic interaction between the PA arch and the PB1 β-hairpin, stabilizing and maintaining the position of the phosphate-binding motif (Appendix A). In fact, introduction of aspartic acid in lieu of the hydrophobic PA arch residues resulted in the suppression of the polymerase activity (Appendix A), further supporting the intolerance of a charge in the hydrophobic core.

Hydrophobic mutants of WSN PB1 were subjected to viral rescue, in which only 362M, 362I and 362L generated infectious virion. 362V and 362A failed to regenerate infectious virion from the plasmid-based viral reconstitution method [25]. This reassured that activity of the polymerase would directly affect the infectivity of the mutant virus. As suggested by our results, 362M and 362L are more favored for virus propagation, as the polymerase retains higher activity and accumulates more mRNA for the translation of viral proteins.

In summary, we have analyzed the PB1 β-hairpin conservation, and inspected the polymerase functions of several naturally occurring PB1 mutants. It was observed that the variant PB1 362L promotes the polymerase activity in both replication and transcription, while other residues reduce the activity. We have also suggested a hydrophobic network between the PB1 β-hairpin motifs and PA arch, which might compensate for the electrostatic interactions between the charged side chains of the two subunits.

## Figures and Tables

**Figure 1 viruses-15-00396-f001:**
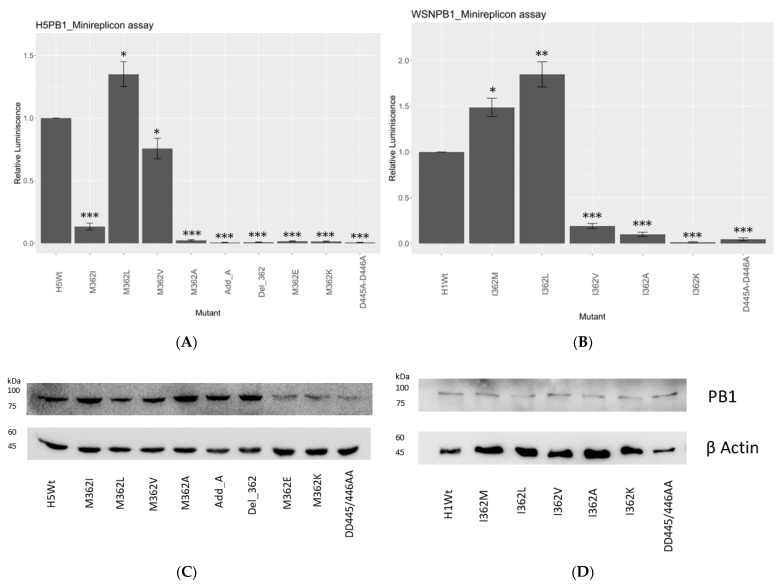
Minireplicon assay with H5N1 and H1N1 RNP. 293T cells were transfected with the plasmids expressing PB2, PA, NP and wild-type or mutant PB1 proteins, along with the FLuc promoter and the GFP control in (**A**) 156/97 background and (**B**) WSN background, respectively. The relative luminescence was determined by the ratio of luciferase/GFP signal, and the results of the mutants were normalized with the wild type in both setups. D445A-D446A mutant acted as the negative control. Error bars indicated the standard error of mean for at least three independent experiments. The *p* value was determined by Student’s *t*-test. (* = *p* value < 0.05; ** = *p* value < 0.01; *** = *p* value < 0.001). Mutants of 156/97 PB1 (**C**) and WSN PB1 (**D**) showed similar expression levels after normalizing to the corresponding β actin controls (Appendix A).

**Figure 2 viruses-15-00396-f002:**
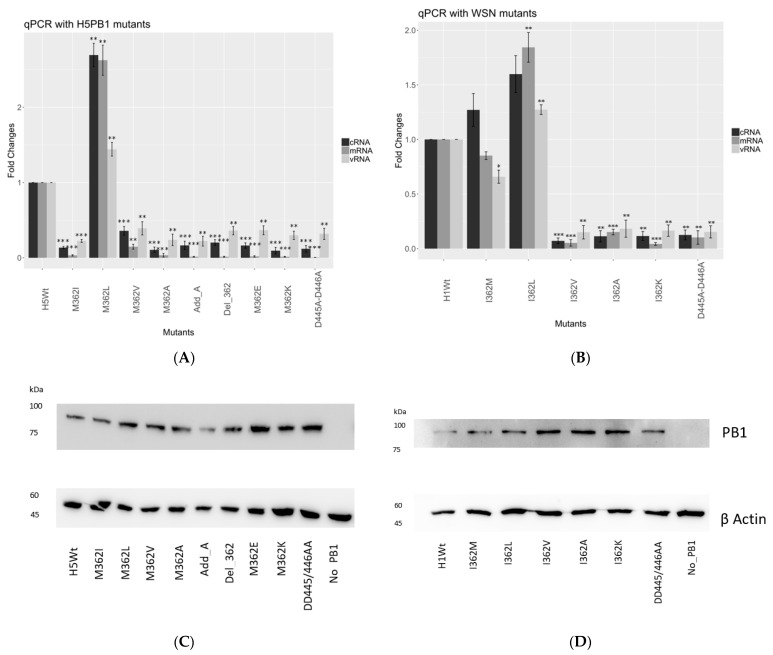
Quantitative PCR for the hydrophobic mutants. Effect of the mutants on the RNA synthesis pathways was determined by qPCR. Total RNA harvested 24 hours post-transfection was reverse transcribed by artificially tagged primers, with 5s rRNA as the internal control. qPCR was performed by the amplification between the tag and segment. (**A**) The relative RNA expression level of WSN PB1 mutants and (**B**) the relative RNA expression level of 156/97 PB1 mutants. Error bar showed the SEM of three independent experiments and statistical analysis was performed by Student’s *t*-test (* = *p* value < 0.05; ** = *p* value < 0.01; *** = *p* value < 0.001). Samples were probed by specific antibody against PB1 and β actin. Mutants of 156/97 PB1 (**C**) and WSN PB1 (**D**) showed similar expression levels after normalizing to the corresponding β actin control (Appendix A).

**Figure 3 viruses-15-00396-f003:**
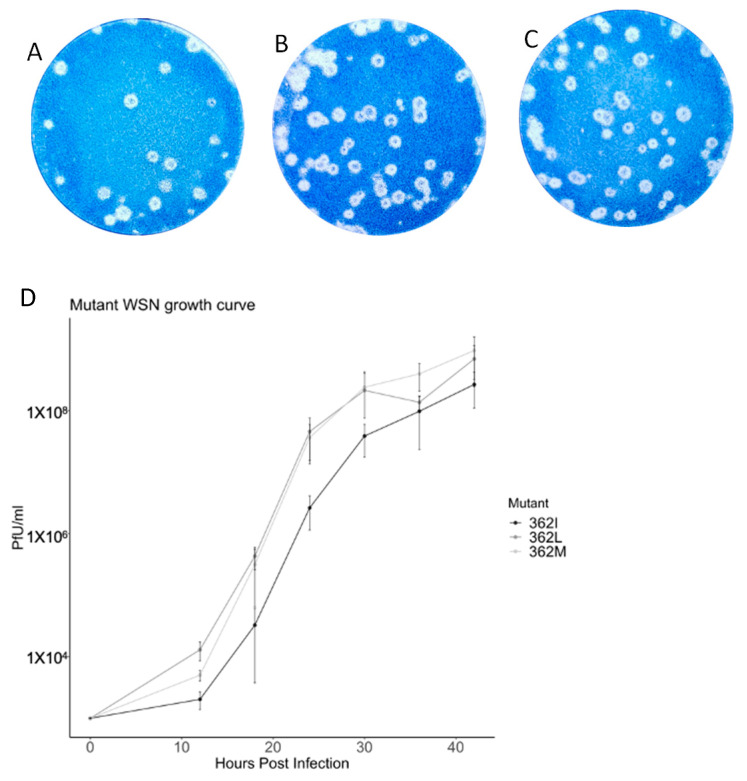
Plaque morphology and growth curves of different WSN mutant viruses. Viruses reconstituted with different PB1_362_ residues at P3 were infected to monolayer of MDCK cells and plaque assay was performed. All of the residues (**A**) 362M, (**B**) 362I and (**C**) 362L with polymerase activity generated infectious virions upon the 3rd passages. Remaining hydrophobic mutants, 362V and 362A, showed no plaque formed upon the 5th passage, suggesting the lack of ability to generate infectious virions. (**D**) Growth kinetics of WSN variants. 362M showed a lower growth rate within the first 12 h post-infection, while similar growth kinetics was observed compared with 362L after 18 h. 362I has the growth rate consistently lower than the two mutants.

**Figure 4 viruses-15-00396-f004:**
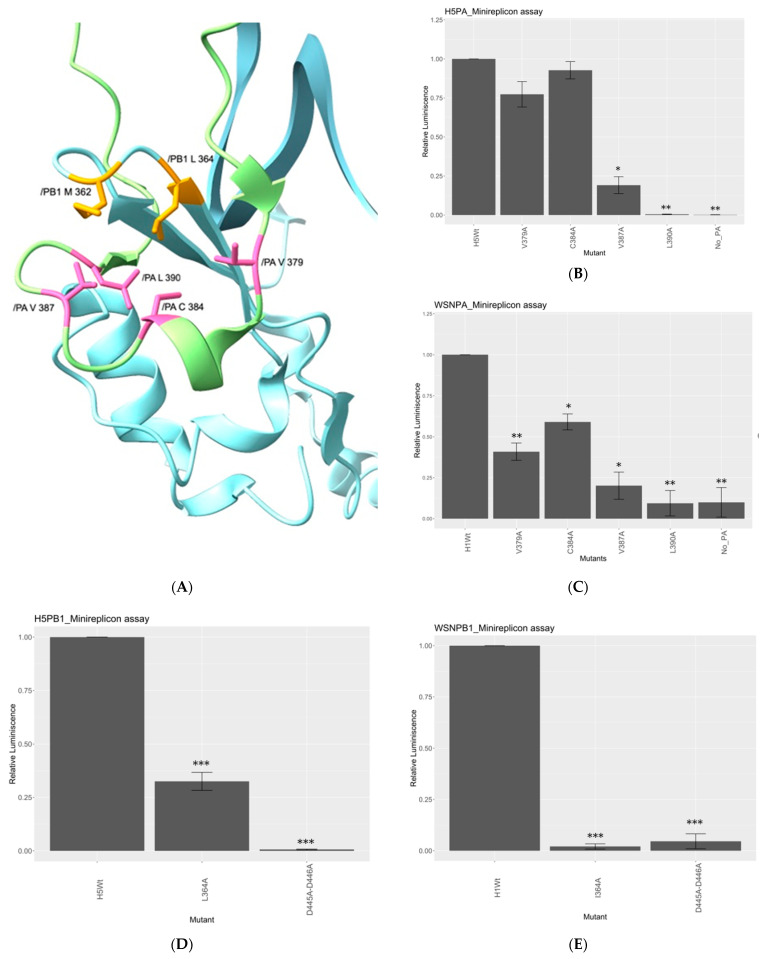
Hydrophobic residues in PA arch interacting with PB1 β-hairpin. (**A**) The proposed hydrophobic network integrated by the PB1 β-hairpin and the PA arch motif, in which the hydrophobic residues on the PA arch were labelled in pink and those in PB1 labelled orange. The side chains of the mentioned residues were pointing towards the center of the network, suggesting the structural importance of the hydrophobic residues within the network (RCSB ID: 6RR7). The effect of PA alanine mutants on the polymerase activity were determined by minireplicon assay. Activity of PA mutants in 156/97 background (**B**) and the WSN background (**C**) showed a similar trend. RNP cassette without PA was used as negative control in both set-ups. The effect of WSN PB1/I364A and 156/97 PB1/L364A mutants were determined by minireplicon assay. Activity of L364A in 156/97 background (**D**) reported an about 70% decrement in polymerase activity, while I364A in WSN background (**E**) defected the RNP activity. D445A-D446A was used as the negative control in both set-ups. Student’s *t*-test was performed to determine the *p*-value. * = *p* value < 0.05; ** = *p* value <0.01; *** = *p* value < 0.001.

## Data Availability

No new data were created or analyzed in this study. Data sharing is not applicable to this article.

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
