# Peer review of "Functional Importance of the Hydrophobic Residue 362 in Influenza A PB1 Subunit"

_viruses, 2023, doi:10.3390/v15020396_

Round 1

Reviewer 1 Report

Comments to the Author

In this report, the authors present functional importance of the hydrophobic residue 362 in PB1 subunit interacting with PA-arch. Different types of mutant (M362I, M362L, M362V, M362 et al.) were subject to the Minireplicon assay and the Western Blot assay to evaluate the RNA-polymerase activity and the PB1 expression, respectively. These results suggested that the M362 is highly conserved in a variety of strains of IAV because it is critical for structural functions between PA-arch-PB1-β-hairpin. This manuscript is important to provide insight into how the hydrophobic pockets consisting of M362 and I or L364 interacts with PA-arch promoting the polymerase activity. Taken together, the manuscript is endowed with sufficient novelty, and could be accepted for publishing in Viruses after minor revision followed the suggestions provided below.

Minor revisions:

(1)In supporting information Fig S1, (A) the β hairpin of PB1 (348-377) PB1 (363) is missing.

   Footnote:  PB1 β hairpin (375-396) PB1 β hairpin (375-395)

(2) Figure 2:  The resolution of images C and D was not fine.

Reviewer 2 Report

Chan et al describe a novel hydrophobic core within PB1 that appears to be necessary for polymerase activity. They show in minireplicon assays that polymerase activity can be increased in two different IAV RNPs by mutating the wildtype PB1 362M to 362L (or I to L in the case of WSN). Additionally, mutations to an acidic residue, addition or deletion of the residue are not tolerated and have similar minigenome activity as an enzymatic knock out mutant.  They also show that WSN has delayed growth kinetics when PB1 362 is mutated from I to M but no impacts are seen with 362 I to L. Based on existing crystal structures they propose a PA-PB1 contact that is dependent on PB1 362, but are only able to show loss of function by PA alanine scanning mutations. While this is a unresearched structural area of IAV PB1, the conclusions that can be made from the work is limited. 

The choice of WSN as the model strain for this proposed interaction is perplexing? If methionine is the dominant residue across 99% of sequenced IAV PB1 why did the authors try to rescue a virus that has already adapted to carry a residue that dramatically impairs activity in the H5N1 minigenome assay (PB1 362I). This conundrum is not addressed by the authors. Additional viral rescue in PR8 (which does utilize PB1 362M) is a necessary control to ensure that WSN is not an aberrant virus in its utilization of this hydrophobic pocket. 

Additionally, the studies within PA do not contribute to this manuscript as they only show loss of function with alanine scanning.  Much of the influenza polymerase loses function with alanine mutagenesis and so lacking any complementary mutations in PB1 that allow for restored function these mutations do not provide any information to the reader on the story of PB1 362. 

Lastly the authors do not correctly interpret their results then they discuss their qPCR results which do not provide any useful information given the width of their error bars. No conclusions can be reached about any differential impacts of mRNA versus vRNA vs cRNA based on this data. 

Round 2

Reviewer 2 Report

The additional replicates support the hypothesis tested in their RT-PCR and the addition of testing charged residues within the proposed PA interface greatly increase the strength of their data.  Overall this does suggest that the hydrophobic interface they propose does exist and is important for polymerase function.